# Bioactive Nitrosylated and Nitrated *N*-(2-hydroxyphenyl)acetamides and Derived Oligomers: An Alternative Pathway to 2-Amidophenol-Derived Phytotoxic Metabolites

**DOI:** 10.3390/molecules27154786

**Published:** 2022-07-26

**Authors:** Sergey Girel, Vadim Schütz, Laurent Bigler, Peter Dörmann, Margot Schulz

**Affiliations:** 1Department of Chemistry, University of Zurich, CH-8057 Zurich, Switzerland; sergey.girel@unige.ch; 2Institute of Molecular Physiology and Biotechnology of Plants, University of Bonn, D-53115 Bonn, Germany; vaschuetz@uni-bonn.de (V.S.); doermann@uni-bonn.de (P.D.)

**Keywords:** nitrated/nitrosylated *o*-acetamido-phenol, oligomers, phenoxazinone, microorganisms, *Aminobacter aminovorans*, *Paenibacillus polymyxa*, *Arthrobacter* MPI764, *Arabidopsis thaliana*, terpene synthase TPS04

## Abstract

Incubation of *Aminobacter aminovorans*, *Paenibacillus polymyxa*, and *Arthrobacter* MPI764 with the microbial 2-benzoxazolinone (BOA)-degradation-product *2*-acetamido-phenol, produced from 2-aminophenol, led to the recently identified *N*-(2-hydroxy-5-nitrophenyl) acetamide, to the hitherto unknown *N-*(2-hydroxy-5-nitrosophenyl)acetamide, and to *N-*(2-hydroxy-3-nitrophenyl)acetamide. As an alternative to the formation of phenoxazinone derived from aminophenol, dimers- and trimers-transformation products have been found. Identification of the compounds was carried out by LC/HRMS and MS/MS and, for the new structure *N-*(2-hydroxy-5-nitrosophenyl)acetamide, additionally by 1D- and 2D-NMR. Incubation of microorganisms, such as the soil bacteria *Pseudomonas laurentiana*, *Arthrobacter* MPI763, the yeast *Papiliotrema baii* and *Pantoea ananatis,* and the plants *Brassica oleracea* var. *gongylodes* L. (kohlrabi) and *Arabidopsis thaliana* Col-0, with *N*-(2-hydroxy-5-nitrophenyl) acetamide, led to its glucoside derivative as a prominent detoxification product; in the case of *Pantoea ananatis*, this was together with the corresponding glucoside succinic acid ester. In contrast, *Actinomucor elegans* consortium synthesized 2-acetamido-4-nitrophenyl sulfate. 1 mM bioactive *N*-(2-hydroxy-5-nitrophenyl) acetamide elicits alterations in the *Arabidopsis thaliana* expression profile of several genes. The most responsive upregulated gene was pathogen-inducible terpene synthase *TPS04.* The bioactivity of the compound is rapidly annihilated by glucosylation.

## 1. Introduction

Benzoxazinones (BXs) are important secondary metabolites characteristically produced in young tissue of several Poaceae, such as maize, wheat, and rye, and some dicotyledonous species [1,2]. The compounds are phytotoxic and possess bioactivity against certain insects, fungi, and bacteria [3,4,5,6]. BXs seem to have additional functions, for instance in phytohormone signaling pathways and in chemotaxis [7,8]. The unstable aglycons of two important BXs, 2,4-dihydroxy-7-methoxy-1,4-benzoxazin-3(4*H*)-one (DIMBOA, from glycosylated DIMBOA) and 2,4-dihydroxy-1,4-benzoxazin-3(4*H*)-one (DIBOA, from glycosylated DIBOA) are degraded to the benzoxazolinones 6-methoxy-benzoxazolinone (MBOA) and benzoxazolin-2(*3H*)-one (BOA), which are still moderately phytotoxic. However, many plants detoxify benzoxazolinones after uptake, either by hydroxylation and subsequent *O*-glucosylation or by *N*-glucosylation yielding glucoside carbamates [9].

To avoid harmful accumulation of the biocides in the soil and in the rhizosphere and to counteract autotoxification of BX-containing plants, detoxification pathways, degradation, or microbial activities suitable to eliminate the compounds are of high importance. At present, only a few microbial detoxification and degradation products of benzoxazolinones are known. Many bacteria are able to nitrate the plant BOA-detoxification intermediate 6-hydroxy-[d]oxazol-2(3H)-one in position 5, yielding biodegradable 6-hydroxy-5-nitrobenzo[d]oxazol-2(3H)-one [10]. *Fusarium* species cleave the benzoxazolinone heterocycle by lactamase activity and form, via the intermediate aminophenol (**1**), the detoxification products acetamidophenol (**2**), and *N*-(2-hydroxyphenyl) malonamic acid (Figure 1) [3,11,12,13]. However, **1** is also the precursor of 2-amino-3H-phenoxazin-3-one (**3**), [14,15]. Compound **3**, generated spontaneously by the dimerization of **1** or by microbial phenoxazinone synthases, is highly toxic to many plants and fungi. Since **3** has a longer lifetime in soil than benzoxazinones and benzoxazolinones, the compound is presumable mainly responsible for the allelopathic effects of young rye plants (*Secale cereale* L.) or of other BX-containing seedlings that release benzoxazinoids into the soil, for instance, by root exudation. The selective fungitoxic properties of **3** may contribute to a shifting in the fungal-species composition, with impacts on the biodiversity in the soil by placing sensitive species at a disadvantage [6].

Insensitive endophytic fungi, isolated from BX containing *Aphelandra tetragona* (Vahl) Nees*,* degrade BOA and HBOA to the known microbial products **1** and **2** and, when cultured in a nitrate rich environment, to *N*-(2-hydroxy-3-nitrophenyl) acetamide (4) and *N*-(2-hydroxy-5-nitrophenyl) acetamide (**5**), [16]. In a recent study [17], **5** was also identified as a compound produced by the soil bacteria *Aminobacter aminovorans* and *Paenibacillus polymyxa*, when the bacteria were cultured in Czapek medium. The nitration of **2** is assumed to activate the compound for subsequent reactions, leading to detoxification and elimination by oligo/polymerization, as was suggested for *A. aminovorans* and *P. polymyxa.*

To shed more light on the bacterial nitration of compound **2** and the identification of its precursors and derived oligomers, in this study we dissected the incubation media of *A. aminovorans* and *P. polymyxa* for hitherto unidentified bacterial-nitrated isomers and subsequent products. Then, we detangle the derived oligomers as precursors for a possible polymer that coats *A. aminovorans* and *P. polymyxa*. Further, we investigated microbial and plant-microbe strategies to cope with **2** and its nitrated derivative **5**, by identifying their conversion products present in microbial incubation media after incubation of **5**. Root extracts of *Brassica oleracea* var. *gongylodes* L. (kohlrabi) incubated with the 4-, 5-, and 5-NO_2_ derivatives of **2** or of *Arabidopsis thaliana* Col-0 plants treated with **5** were also examined. Finally, the possible bioactive properties of **5** were investigated by studying the expression of defined *Arabidopsis* genes to get first clues for the compound’s involvement in modulating gene activities, which were, however, found to be quickly attenuated or blocked by efficient microbial or plant glucosylation.

## 2. Results and Discussion

### 2.1. Synthesis, Purification, and Identification of Compounds

#### 2.1.1. Identification of the Nitrated Derivatives of 2-Acetamido-phenol

*Aminobacter aminovorans* and *Paenibacillus polymyxa* culture media supplemented with **2** contained an accumulation of several new compounds after one week of incubation. The culture media were partitioned with EtOAc, and the aqueous phases were collected, evaporated to dryness, and analyzed with liquid chromatography coupled to high-resolution mass spectrometry (LC/HRMS), revealing the presence of four major compounds in the aqueous phase reconstituted from the culture medium of *Aminobacter aminovorans* (*F49*, Figure 2, peaks *A1-A4*). The analysis of the aqueous phase of the *Paenibacillus polymyxa* medium (*F51*) resulted in a similar chromatogram, featuring four major peaks with comparable retention times and masses in both (+)- and (–)-ionization modes (Appendix A). Due to the similarity of the two bacterial-culture extracts, *F49* was used for structure elucidation of the four components, while *F51* was considered as backup because of the small amount of material available and its susceptibility to degradation.

The four major components of the aqueous phase *F49* were identified as the substrate **2** (*A1*, commercially available), the nitroso derivative **6** (*A2*, structure elucidation shown below), and the two nitro derivatives **4** (*A4*, synthetic reference material, Table 1) and **5** (*A3*, commercially available).

#### 2.1.2. Identification of *N*-(2-hydroxy-3-nitrophenyl)acetamide in the Fraction A4 of *P. polymyxa* Bacterial Culture Extract

To assess the structure of the compound corresponding to the LC/MS peak *A4*, the culture medium of *P. polymyxa* was fractionated on eight separate runs with an HPLC-UV system equipped with a 4.6 mm core shell column (Appendix A). The peaks of interest, *P1*-*P4*, were collected and pooled, and the solvent was evaporated (see exp. part). The NMR characterization of the fraction *P4* residue failed due to its decomposition during lyophilization. Fortunately, ESI-HR-MS/MS analyses (Appendix A) allowed the identification of the metabolite as *N-*(2-hydroxy-3-nitrophenyl)acetamide (**4**), by comparison with the fully characterized synthetic reference material (synthesis described in Material and Methods and NMR data in Appendix A). Furthermore, all spectra were in agreement with published data [16]. Finally, the presence of two remaining positional isomers *N-*(2-hydroxy-6-nitrophenyl)acetamide (**7**) and *N*-(2-hydroxy-4-nitrophenyl)acetamide (**8**) in the bacterial extract was excluded by a comparative LC/MS analysis and diverging retention times (Appendix A).

#### 2.1.3. Identification of *N*-(2-hydroxy-5-nitrosophenyl)acetamide in the Fraction A2 of *P. polymyxa* Bacterial Culture Extract

The identification of the compound contained in fraction *A2* was more difficult, as it has not been described before. The atomic composition of C_8_H_8_N_2_O_3_ was deduced from the exact masses *m*/*z* 179.04637 ([M–H]^–^, C_8_H_7_N_2_O_3_^−^, +0.2 mDa) and 181.06028 ([M+H]^+^, C_8_H_9_N_2_O_3_^+^, −0.5 mDa) observed in the full-scan mass spectra (Appendix A). Possible derivatives of 2-acetamido-phenol featuring this chemical form are *o*-, *m*-, and *p*-nitroacetanilides, on the one hand, and 2-hydroxy-nitrosophenyl-acetamides, on the other hand (with nitroso-substituent at position 3-, 4-, 5-, or 6).

The structure elucidation was achieved with the (+)-ESI-MS/MS (Figure 3) and NMR (Appendix A) of the purified fraction *P2* of the extract. The fragmentation of the unknown precursor *m*/*z* 181.06028 ([*M*+H]^+^) revealed a major fragment ion at *m*/*z* 109.05257 (100% int, C_6_H_7_NO^•+^, +0.4 mDa), accompanied by minor fragments at *m*/*z* 139.05023 (6%, C_6_H_7_N_2_O_2_^+^, <0.1 mDa), 95.03709 (5%, C_5_H_5_NO^•+^, +0.5 mDa), and 80.05015 (10%, C_5_H_6_N^+^, +0.7 mDa). The common C_2_H_2_O neutral loss is facilitated by an ortho-effect from a neighboring OH functionality and indicates the presence of an N-acetyl group. This cleavage is also observed in the MS/MS spectra of the commercially available compounds **2** and **5** (Figure 3).

The presence of the acetamide moiety that was proposed from the MS/MS data (see above) is confirmed with the singlet at 2.15 ppm (3H, C*H_3_*) in the ^1^H-NMR spectrum of the fraction *P2* (Appendix A). Furthermore, three aromatic protons containing two doublets at 7.38 and 6.55 ppm and a singlet at 8.40 ppm are detected. Such an arrangement immediately ruled out nitro-acetanilides as they presuppose a disubstituted phenyl ring. Nitroso substitutions at position 3 and 6 were also found to be inappropriate because the ^1^H-NMR featured one isolated and two protons coupling with each other (^1^H,^1^H-TOCSY in Appendix A). Comparison with 1D- and 2D-NMR spectra obtained from commercial reference material **5** (data not shown) and from synthetic compound **4** allowed for a secure assignment of all remaining signals at 13.56 ppm (*brs* 1H, O*H*) and 9.40 ppm (*brs* 1H, N*H*), respectively. The exact position (4 or 5) of the nitroso group was clarified by estimating the ^1^H-NMR chemical shift of the corresponding singlet with ChemDraw (vers. 21.0.0.28, Perkin Elmer Informatics, Inc.). Comparison of the calculated values of 7.27 (4-NO) and 8.04 ppm (5-NO) with the experimental value of 8.40 ppm clearly indicates the presence of the nitroso group in position 5.

Additional evidence for the structure of compound **6** was obtained from an LC-(+)-ESI-is CID/HR-MS^2^ analysis of the bacterial extract *F49*, where *in-source* generated fragments of precursor ions *m*/*z* 181.06038 (peak *A2*) were isolated and MS/MS spectra of each fragment were recorded and compared to the fragmentation behavior of peak *A3* (**5**). Both compounds first lost ethenone to produce the corresponding nitroso (*m*/*z* 139.04990) or nitrophenol (*m*/*z* 155.04489) fragments. In a subsequent cleavage, losses of ^•^NO (**6)** or ^•^NO_2_ (**5**) radicals were observed, yielding the 2-aminophenol radical ion (Appendix A). Radical cleavage of aromatic nitro and nitroso derivatives have been described recently [21]. To our best knowledge, this is the first description of the proposed structure of *N-*(2-hydroxy-5-nitrosophenyl)acetamide (**6**).

#### 2.1.4. Presence of Dimers and Trimers Derived from 2-acetamido-phenol

Next, oligomers of compounds **2**, **5**, and **6** and their nitrated and nitrosylated derivatives were addressed with LC/HRMS. Since most of them were found only in low amounts, high-resolution extracted ion chromatograms (calculated from the mass corresponding to the chemical form of 2-acetamido-phenol dimers (±1.5 mDa) and trimers (±2 mDa) have been used for improved sensitivity. The analysis of the total *A. aminovorans* culture extract *F49* revealed a series of dimers (D1–D11), as shown in Figure 4A. Four dimers (EIC *m*/*z* 299.10373), eluting at 3.36, 5.64, 6.71, and 6.97 min, four nitrosylated dimers (*m*/*z* 328.09389, 5.26, 5.71, 6.26, and 8.12 min), and three nitrated dimers (*m*/*z* 344.08826, 8.14, 8.80, and 9.09 min). A second series corresponding to trimers (*T1*–*T16*) was dissected in the 12.5 min fraction obtained from the preparative purification of *F49* (Appendix A). Six trimers (*m*/*z* 498.15141), with *T4* (*t_R_* 7.69 min) as the major peak, five nitrosylated trimers (*m*/*z* 477.14157) with the most abundant metabolite *T10* at 7.29 min, and five nitrated trimers (*m*/*z* 493.13649), again containing a major component (*T16*, 9.14 min), were detected. The classification of the metabolites was confirmed by MS/MS experiments (Appendix A), but the exact molecular structures of their dimer and trimer isomers could not be resolved as there was not enough material available for NMR experiments after isolation and purification. The general structure of the dimers, trimers, and the corresponding peaks are shown in Figure 4 and listed in Appendix A. Oligomers (dimers, trimers, and a tetramer) are known to be formed from 4-acetamido-phenol.The 4-acetamidophenol oligomers were produced by horseradish peroxidase reaction in presence of H_2_O_2_ [22]. To our knowledge, oligomers of 2-acetamido-phenol have not been described so far.

### 2.2. Bacterial and Fungal Detoxification Products Derived from N-(2-hydroxy-5-nitrophenyl)acetamide

As shown, *A. aminovorans* and *P. polymyxa* synthesize 2-acetamido-phenol-derived oligomers, in contrast to other microorganisms tested in a prior study [17]. In nature, linking precursor monomers to oligomers or conjugation of precursors with primary metabolites, such as sugars, presents efficient methods to eliminate or to reduce amounts of bioactive molecules. In the following set of experiments, we aimed to elucidate if further microorganisms are able to use *N*-(2-hydroxy-5-nitrophenyl)acetamide for oligomer production. For this purpose, strains of the bacteria *Bacillus amyloliquefaciens, Pantoea ananatis*, *Pseudomonas laurentiana*, *Pseudomonas* spec. *MPI 9*, and *Arthrobacter* (two species), the yeast *Papiliotrema baii*, and consortia of *Actinomucor elegans* and *Trichoderma viride* were incubated with substrate **5**. We also included *A. aminovorans* and *P. polymyxa*, to check whether these bacteria can start oligomer production directly from compound **5**. *Pseudomonas spec*. MPI 9 and the two *Arthrobacter* species were also incubated with substate **2**. LC-ESI-HRMS analysis of the aqueous phases of the cultures revealed that the organisms handle the nitrated product **5** differently (Figure 5).

*Pantoea ananatis*, *Pseudomonas laurentiana*, and the yeast *Papiliotrema baii* glucosylated *N*-(2-hydroxy-5-nitrophenyl)acetamide to conjugate **9***. P. ananatis* produced an additional compound, which was tentatively identified as a succinate derivative of **9** (**10**, Appendix A). *Bacillus amyloliquefaciens* was neither able to produce a glucoside derivative, nor to accumulate any other product. The *Actinomucor elegans* consortium did not glycosylate **5**, although *P. ananatis* and *P. baii* are hosted within the nexus of the hyphae. Albeit, though not proven, the simplest explanation could be a protection of *P. ananatis* and *P. baii* by the cotton-like mycelium, shielding the associated organisms from facing substrate **5** and, consequently, preventing glycosylation. Instead, the sulfated product **11** was detected as the only *N*-(2-hydroxy-5-nitrophenyl)acetamide derivative (structure shown in Figure 5). We assume that the sulfation is performed by the Zygomycete as the dominate organism of the consortium. The same product was found in trace amounts and as the only conversion product in the samples obtained from *Trichoderma viride* consortium incubated with **5**. Thus, the two consortia convert **5** via a pathway similar to *Phase 2* metabolism in mammals. Accurate MSMS spectra and structure elucidation of compounds **9**, **10**, and **11** are presented in Appendix A, and the corresponding data are summarized in Appendix A.

The *Pseudomonas* species MPI 9 did not metabolize both compounds. The *Arthrobacter s*pecies MPI 763 glycosylated **5**, while 2-acetamido-phenol was not converted. Thus, many bacteria detoxify **5** by glucosylation. However, the resulting **9** might not always be persistent. When *P. ananatis* and *P. laurentiana* were incubated with **9**, the glucoside was hydrolyzed again, while **5** accumulated (Appendix A).

In reference to the identified nitro- and nitroso-compounds, *Arthrobacter* species MPI 764 metabolized 2-acetamido-phenol, yielding **4**, **5**, and **6** in very low amounts (Appendix A). The experiment with *Arthrobacter* species MPI 764 indicated that the ability to synthesize these compounds from **2** is not restricted to *A. aminovorans* and *P. polymyxa*. The incubation study with **5** disclosed, however, that neither *A. aminovorans* and *P. polymyxa*, nor *Arthrobacter* species MPI 764, could synthesize the dimers and trimer when the nitro derivative **5** was offered instead of **2**. It is conceivable that **5** is a starter molecule for the formation of linked 2-acetamido-phenol molecules, so the nitro group is subsequently removed. We assume a necessity of **5** and **2** in co-existence for oligomer production, but biosynthetic steps and enzymes involved are presently not known. A channeling of the synthesis, starting with substrate **2**, cannot be excluded. In contrast to the formation of 2-amino-3*H*-phenoxazin-3-one (**3**) from 2-aminophenol (**1**), it seems that the alternative pathway occurs under low-oxygen conditions.

The nature of the products depends on the culture conditions. When *A. aminovorans* and *P. polymyxa* cultures supplemented with 2-acetamido-phenol are strongly shuttled during incubation, none of the above-described nitro compounds could be identified. Instead, metabolite **3** and a related, not-further-analyzed derivative were found in the medium. The occurrence of **3** in the medium is only possible when the bacteria de-acetylate **2**, yielding 2-aminophenol, which should be a prerequisite for the catabolic breakdown of the aromatic system. Cessation of shuttling, at the latest two days after adding substrate **2**, led to the detection of some of the conversion product **3** but also **4**, **5**, and **6** (Figure 6). Consistently, the accumulation of **4**, **5**, and **6** versus the production of **3** probably depends on the oxygen availability in the medium, which is low without shuttling, and on the physiological conditions of the bacteria. The assumption needs further investigation, as oxygen thresholds for pathway switches could be species-dependent. The identification of the compounds was performed with the commercially available compounds **4** and **5**, and with the synthetic reference materials **3** (catalytic cyclo condensation) and **6** (see Exp. Part) [6].

### 2.3. Incubation of Plants with Nitrated Derivatives of 2-Acetamido-phenol

We went further into the question of how plants cope with natural *N*-(2-hydroxy-5-nitrophenyl)acetamide (**5**). For this purpose, we chose two Brassicaceae, the legume *Brassica oleracea* var. *gongylodes* L. (kohlrabi), and *Arabidopsis thaliana* Col-0. Kohlrabi was also incubated with synthetic *N*-(2-hydroxy-6-nitrophenyl)acetamide (**7**) and *N*-(2-hydroxy-4-nitrophenyl)acetamide (**8**). After exposure of kohlrabi seedlings to **5** and **7**, the methanolic extracts of the roots contained several metabolites (Figure 7, annotation in Appendix A), whereas **8** was not converted. Compound **7** was glucosylated in high yields at the 2-*O*-position to give **15**. Compound **15**, a derivative of **7** that exhibits acetylation of the glucosyl substituent, was further detected as a minor metabolite. A small amount of **7** was deacetylated to yield 2-amino-3-nitrophenol (**18**). Then, **5** was glucosylated at the 2-*O*-position yielding compound **9**, the same one as found with *P. ananatis*, *P. laurentiana*, *Arthrobacter s*pecies MPI 763, and the yeast *P. baii.* Moreover, the incubation of three-week-old *Arabidopsis thaliana* plants with **5** resulted in high amount of the glucosylation product **9** (Appendix A), primarily in the roots.

The finding that the roots of kohlrabi accumulated an unexpected high amount of glucosylated compound **9** indicates that associated microorganisms are able to perform the glucosylation step and supports the role of this plant species in compound deactivation. In this context, we suggest that the modified form of N-(2-hydroxy-5-nitro- phenyl)acetamide glucoside succinate (**10**) produced by *Pantoea ananatis* is somewhat resistant to enzymatic hydrolysis.

#### 2.3.1. Effects of *N*-(2-hydroxy-5-nitrophenyl)acetamide on Arabidopsis Thaliana Phenotype and Accumulation of *N*-(2-hydroxy-5-nitrophenyl)acetamide Glucoside

In contrast to nitrated BOA-6-OH [10], which causes yellowish spots on *Arabidopsis* leaves and the rolling up of leaf margins when exposed at 0.5 mM concentrations, substance **5** had no comparable effects up to 1 mM, even after a 48-h incubation (Figure 8). Higher concentrations, e.g., 2 mM and more, however, led to the wilting of young leaves (Figure 8). Incubations with 1 mM of compound **5** revealed the accumulation of the substate in the seedlings at a concentration of ~30–40 nmol/g fresh weight (FW) after 30 min. After 1 h, ~40–50 nmol/g FW were found together with traces (<1 nmol) of the glucoside **9**. The glucoside increased to averaged 9 nmol/g FW after 3 h, whereas the aglycon remained almost at the 1 h level, then dropped to about 20–25 nmol during the next 2 h (5 h exposure), concomitant with an increase in the glucoside to 13–15 nmol/g FW. After 24 h, about 60 nmol of **9** accumulated, while the aglycon further dropped to trace amounts. Overall, the amounts of absorbed **5** and produced **9** were low, and intermediate derivatives of even lower abundance present in the root extracts could not be identified. In the incubation medium, **9** did not accumulate nor was there any other stable conversion product in verifiable amounts. Since *Arabidopsis* plants still harbor microorganisms, despite seed-surface sterilization [23], the glucosylation step can be performed by plant glucosyltransferases but also by competent endophytic microorganisms. Glucosylation masks the bioactivity of the compound, which increasingly counteracts the aglycon′s effects on the plant. Promiscuous bacterial glucosyltransferases able to glycosylate various secondary metabolites are known, for instance, glucosyltransferase UGT102B1 from *Pantoea ananatis* or UGTs from the *Pseudomonas* species. While emphasizing the molecular-structure elucidation, the identification of the responsible glucosyltransferases was not addressed in this study.

#### 2.3.2. *N*-(2-hydroxy-5-nitrophenyl)acetamide Influences Gene Expression

Nitroaromatic compounds are known for their biological activity, often disclosed by microbial reductive metabolization steps, for instance, by anaerobic reduction via nitroso- and hydroxylamine intermediates. Actually, the intermediates are thought to be important bioactive compounds. The first steps of a reductive metabolization can also lead to the release of nitrogen oxide species, such as nitric oxide (NO), nitrite (NO_2_^−^), or HNO and others. Further, hydroxylamine or ammonium can be released. By use of synthetic nitro aromatic compounds, microbial nitroreductases, oxireductases, and hydrogenases have been identified as enzymes involved in reductive-pathway steps [24,25]. Since we identified the nitroso aromatic compound **6** as a bacterial product, the question was raised if **5**, via **6**, can serve as a source for nitric oxide/nitrogen oxide species release by the metabolic activities of plants or plant-associated microorganisms. In plants, NO reservoir molecules are known, with nitroso glutathione (GSNO) and nitro linolenic acid as prominent examples [26].

In plants, nitric oxide at concentrations in the pmol and low nmol range is an important signal molecule with crucial functions in metabolism and development, in lateral root formation, in stress reactions including plant–herbivore interaction, in plant immunity, and in others [26,27]. Generally, NO has a genome-wide influence on gene expression [28,29]. NO affects transcription, for instance of the terpene synthases (*TPS04* and *TPS02)*, tryptophan synthase *TRYPS02*, and tocopherol cyclase *VTE1*.

Several possibilities for NO generation exist in plants. One important way is the nitrate reduction by nitrate reductases in the cytosol and at the plasma membrane, but NO can be produced also non-enzymatically in the acidic environment of the apoplast, or it can be released from microorganisms [30]. Two cytosolic nitrate reductases exist in *Arabidopsis*, *NIA1* and *NIA2*, whereby *NIA2* is responsible for the reduction in *NIA1*-produced nitrite to NO [26,31,32]. To get clues for an interference of **5** with plant-nitrate reduction, the expression profiles of *NIA1* and *NIA2* were studied. Further, we chose several genes known to be NO responsive [29]. We took aim at *TPS04* and *TPS02* (terpene synthases 04 and 02, both NO-upregulated) as well as tryptophan synthase *TRYPS02* (NO-upregulated) and tocopherol cyclase *VTE1* (NO-down regulated). VTE1 is a key enzyme in tocopherol synthesis [33]. Since deamination and the release of ammonium might alternatively occur (see above), the genes of two cytosolic glutamine synthases *(GLN1.1* and *GLN 1.2*) were included because of their functions in ammonia assimilation and scavenging. *GLN1.1* is known for its high affinity for ammonium (K_m_ < 10 µM), whereas the one of *GLN1.2* is low (K_m_ 2450 ± 150 µM) [34]. Both glutamine synthases are major isoforms in the young plant [35].

qPCR studies revealed that *TPS04* was the most responsive gene after incubation with substrate **5**, which was upregulated more than 4.6-fold after 30 min and 2.8-fold after 1 h of incubation. Subsequently, the relative abundance of transcripts decreased again, and, after 24 h, a more than 3.3-fold downregulation was demonstrated (Figure 9). Interestingly, the geranyllinalool synthase encoding *TPS04* gene was described to be upregulated by the moth *Plutella xylostella*, by the fungal pathogens *Botrytis cinerea* and *Alternaria brassicicola*, and by *Pseudomonas syringae macilicola* [36]. The enzyme´s product geranyllinalool and several of its downstream derivatives are known to have inhibitory functions in plant and animal sphingolipid metabolism [37]. Geranyllinalool, for instance, is a potent inhibitor of serine *C*-palmitoyltransferase involved in ceramide synthesis. The fast glucosylation of the elicitor **5** is, therefore, essential for the plant and perhaps for the sphingolipid-synthesizing groups of bacteria and fungi to prevent enhanced geranyllinalool biosynthesis. In contrast, the terpene synthase gene *TPS02* was not responsive to **5** at any time.

The tocopherol cyclase gene *VTE1* was 1.61-fold upregulated after 30 min and 1.4-fold upregulated after 1 h, and, finally, marginally downregulated after 24 h (-0.4-fold). The early upregulation may increase the tocopherol level during the first h of incubation. Interestingly, γ-tocopherol serves as a NO scavenger, when converted to 5-nitro-γ-tocopherol [38]. *TRYPS02* showed no significant response but possessed the tendency for a weak upregulation. *NIA1* and *NIA2* showed no clear response, while *GLN1.1* and *GLN1.2* were slightly downregulated. From these results, we conclude that **5** has a moderate negative influence on *GLN1.1* and *GLN1.2* expression and almost no effect on the tested *NIA* genes. Regarding the *GLN* and *NIA* expressions, no clues of ammonium release were obtained during the course of exposure to compound **5**. Overall, the responsiveness of the studied genes was moderate or low, except for *TRS04*. Here, decreasing transcript levels were concomitant with the start of bacterial and/or plant-dependent glucosylation of **5** that overrides the bioactive properties of the compound (Figure 8 and Figure 9). The true effect on the gene expression is, therefore, concealed. Aside from a possible NO-triggered change of gene expression, a direct interaction of 5-NO_2_/NO-acetamido-phenol derivatives cannot be excluded. Moreover, numerous additional genes, which could not be tested here, may respond to the compound(s). The responsiveness of the microbial genes was not addressed and remains elusive at present.

### 2.4. Implication of Nitration of 2-Acetamido-phenol

Uptake of secondary metabolites in the original form or in a derived one by plants is a known phenomenon and a principle of allelopathy. While the transfer from plant to plant or the uptake of such compounds from soil has been addressed, for instance, for benzoxazinoids, salicylic acid, pyrrolizidine alkaloids, and other compounds [9,39,40], the degradation and detoxification strategies, by involving microorganisms, are less investigated. Particularly, the contribution of bacteria in glucosylation steps for detoxification is underestimated. The complex interaction between plants and microorganisms results in derivatives with a completely different bioactivity or an annihilated one. Since special microbial species are attracted by defined secondary metabolites, plants can hypothetically modulate their microbiomes by uptake of secondary metabolites from neighbored plants, followed by later release, thereby attracting microorganisms that prefer the compounds as a nutrient source [41]. As a consequence, already-associated microorganisms could get into a competitive situation, thus, the elimination and inactivation of foreign bioactive metabolites is an important feature for maintenance of their own niche. On the other hand, due to secondary compound exchange, the microbiomes of neighbored plants can become adjusted to each other to a certain degree.

The species diversity and composition of soil microorganisms, as well as the availability of oxygen and nitrate, direct the further fate of the microbial BOA-degradation product 2-acetamido-phenol, either to phytotoxic phenoxazinones after de-acetylation, which is extractable from suitable soil samples, or to bioactive nitro/nitroso aromatic acetamido-phenol compounds, which are not yet detected in soils. Oligo- and possible polymerization of the nitro compounds impede the detection of monomers in soil samples. Oligo- and polymers can potentially undergo mineralization and contribute to soil fertility and carbon fixation. Figure 10 summarizes the new pathways presented in this study.

Recently, anaerobic microsites, occurring also in well-drained soils, have been recognized as important for soil carbon persistence because they reduce microbial CO_2_ release to the atmosphere [42]. Moreover, high temperatures due to climate change generate hypoxic soil conditions [43]. Thus, under field conditions, anaerobic microsites of soils, heavily loaded with nitrate and organic material [44], are suitable sites for the microbial synthesis of the nitro/nitroso compounds and the resulting polymers, especially under high-temperature conditions. The alternative pathway for 2-aminophenol derived nitro-acetamido-phenol compounds reduces the formation of phenoxazinone, a compound with herbicidal potential.

Our study underlines that environmental conditions influence the detoxification and degradation of secondary metabolites, here leading to known and hitherto unknown compounds with changed bioactive properties. Although the fast glucosylation of absorbed *N*-(2-hydroxy-5-nitrophenyl)acetamide was impeding, and proof of **5** as a NO reservoir molecule is still pending, we obtained clues for the bioactive character of the compound as it influenced the transcriptional activities of *TRS04*. Our study underpins the importance of microorganisms in modulating the bioactivity of plant secondary metabolites and their degradation products by subsequent pathways and by modulating molecular structures of bioactive intermediates that accumulate during degradation pathways. We agree with Madhusoodanan [45] that the microbial regulation of a compound´s bioactivity by modulation of its structure is widespread in nature.

However, *Pseudomonas* species MPI 9 and *Bacillus amyloliquefaciens* were recognized as two microorganisms that did not convert the applied compounds (here, **2** or **5**), a behavior that was previously found with numerous BOA-incubated bacteria [17]. This phenomenon is most likely explainable by the existence of efficient efflux pumps that are able to export absorbed small molecules in an unchanged form. Those efflux transporters have been recognized as highly important in bacterial antibiotic and, interestingly, also in herbicide (glyphosate) resistance [46,47]. Horizontal gene transfer, for instance, via plasmids, contributes to a rapid distribution of this avoidance strategy within a bacterial community. Additional functions of some transporters are emerging, such as a participation in detoxification reactions. In contrast to oligo-/polymerization, detoxification via conjugation with primary metabolites are often, as shown in this study, reversible. A fatal consequence of microbial efflux activities and reversibility of detoxification reactions can be an increased accumulation and a dramatic extension of the lifetime of compounds in ecosystems.

## 3. Materials and Methods

### 3.1. Reference Substances and Compounds Used for Incubations

The 2-Acetamido-phenol (**2**), 2-amino-3-nitrophenol (**18**), and *N*-(2-hydroxy-5-nitrophenyl)acetamide (**5**) were purchased from FluoroChem Ltd., Glossop, UK; *N*-(2-hydroxy-4-nitrophenyl)acetamide (**8**) was from Accela ChemBio Inc., San Diego, CA, USA. *N*-(2-hydroxy-3-nitrophenyl)acetamide (**4**) and *N*-(2-hydroxy-6-nitrophenyl)acetamide (**7**) were synthesized (see below). The 2-acetamido-phenol used for the incubation of microorganisms was purchased from Sigma-Merck, Taufkirchen, Germany. Solvents and reagents for synthesis were from Merck, if not stated otherwise. Sources of other chemicals are mentioned in the text. Glucosylated **5** was extracted Arabidopsis roots incubated with **5**, as described below. Purchased, synthesized and identified isolated compounds were used as references.

### 3.2. Purification of Aqueous Phase F49 (A. aminovorans) for NMR Measurements

In order to obtain material suitable for structure elucidation with NMR spectroscopy, the aqueous phase *F49* was purified with an HPLC separation method based on a *Waters Cortecs C18* 4.6 × 150 mm, 2.7 µm particle size column. Core shell columns were previously reported as effective small scale preparative separation method [29,30,31]. The four subsequently obtained fractions *P1-P4* (Appendix A) corresponded to the four major LC/HRMS peaks *A1-A4* observed at analytical scale (Figure 2).

### 3.3. Synthesis of N-(2-hydroxy-3-nitrophenyl)acetamide (***4***)

A suspension of 0.99 mmol **2** in 10 mL of H_2_O/AcOH mixture (1:1 *v*/*v*) was cooled to 0 °C using an ice bath. Then, 5.73 mmol of 65% HNO_3_ (0.4 mL) were carefully added dropwise under vigorous stirring, immediately turning the reaction mixture to dark red. After 15 min of stirring at 0 °C, additional 5 mL of H_2_O were added, and the reaction mixture was filtered through a silica plug (h = 3 cm, Ø = 2.5 cm). The first filtrate was discarded, and the resulting sludge atop the silica layer was carefully eluted with CH_2_Cl_2_, resulting in the separation of the yellow and reddish layers. The yellow fraction was collected, and the solvent was removed *in vacuo*, yielding a yellow residue of **4** (11.3 mg, 0.058 mmol, 5.9%). N_2_-degassed DMSO-d_6_ (0.7 mL) was immediately added to the flask to avoid decomposition on contact with air. Analytical data: yellow crystals, impure. ^1^H-NMR (600 MHz, DMSO-d_6_): 9.69 (*brs*., NH); 7.98 (*d*, J = 8, arom. H); 7.68–7.66 (*dd*, J = 8.4, 1.6, arom. H); 6.94–6.91 (*t*, J = 8, arom. H); 2.13 (*s*, CH_3_) (conform to [16]. ^13^C-NMR (150 MHz, DMSO-d_6_): 169.5 (*s*, N–C=O); 145.1 (*s*, ar. quat. C–OH); 137.0 (*s*, ar. quat. C–NO_2_); 129.8 (*s*, ar. quat. C–NCOCH_3_); 127.7 (*d*, ar. HC–C–NCOCH_3_); 120.0 (*d*, ar. HC–C–NO_2_); 118.2 (*d*, arom. C–CH–C); 23.6 (*q*, CH_3_), conform to [20]). (+)-ESI HRMS: 197.05530 (100; [M+H]^+^; C_8_H_9_N_2_O_4_, calc. 197.05568, Δ = –0.4 mDa); 155.04486 (42; [M–C_2_H_2_O]^+^ C_6_H_7_N_2_O_3_, calc. 155.04512, Δ = –0.3 mDa).

### 3.4. Synthesis of N-(2-hydroxy-6-nitrophenyl)acetamide (***7***)

The procedure of Crooks et al. (1979) was adapted [48]. A mixture of 49.5 mmol Ac_2_O (5 g), 7.14 mmol 2-amino-3-nitrophenol (1.1 g), and 15 mL CH_2_Cl_2_ was stirred for 5 min until the color turned to deep red. Then, 10 mmol Et_3_N was slowly added. After 24 h of stirring at ambient temperature, 20 mL H_2_O were added to hydrolyze the excess of Ac_2_O, and the water phase was extracted twice with 20 mL CH_2_Cl_2_. Combined organic phases were washed with 1M NaHCO_3_ (2 × 20 mL), 1M NH_4_Cl (50 mL), and sat. NaCl_aq_ (50 mL), then dried using anhydrous MgSO_4_. Evaporation of the solvent *in vacuo* yielded a dark orange residue (1924 mg), which consisted of the mono- and diacetylated products according to LC/HRMS data. This residue was dissolved in 50 mL MeOH and stirred with 9 mmol anhydrous K_2_CO_3_ at ambient temperature until diacetylated product completely disappeared (approx. 3 h). The solvent was removed in vacuo and the residue was dissolved in a small amount of CH_2_Cl_2_, followed by addition of 50 mL H_2_O and acidification with 1M HCl to pH 4. The water phase was extracted with 5 × 30 mL EtOAc. The combined organic phases were washed with 3 × 30 mL brine, dried with anhydrous MgSO_4_, and the resulting solution was filtered through celite. Evaporation of the solvent in vacuo yielded orange crystals of **8** (1193 mg, 6.1 mmol, 85%), which were immediately stored under argon atmosphere and used without further purification. Analytical data: orange crystals, >90% pure (LC/HRMS), decomposes on contact with air. TLC: Rf = 0.21 (hexane/EtOAc, 1:1), (–)-ESI HRMS: 195.04123 (100; [M–H]^–^; C_8_H_7_N_2_O_4_, calc. 195.04113, Δ = +0.1 mDa); 153.03069 (15; [M–C_2_H_2_O]^−^; C_6_H_5_N_2_O_2_, calc. 153.03057, Δ = +0.1 mDa).

### 3.5. Microorganisms and Plants

#### 3.5.1. Microorganisms

The bacteria *Aminobacter aminovorans* (accession OK376288), *Paenibacillus polymyxa* (accession OK376289)*, and Bacillus amylolique-faciens* (accession OK376290) were collected from soil in a previous study [49]. *Pantoea ananatis* (DSM ID 14-714C) and the yeast *Papiliotrema baii* (DSM 100638, preserved in the open collection of the Leibniz Institute DSMZ, Braunschweig, Germany) were isolated from an *Actinomucor elegans* consortium; the latter was also used entirely for incubations [12,50]. The Zygomycete *Actinomucor elegans,* which incorporated *P. ananatis* and *P. baii* during an inoculation experiment from *Abutilon* roots, was formerly identified (accession KM404167), [50]. The fungus was further identified by its morphology and is registered at the DMSZ as strain AbRoF1. The *Trichoderma viride* voucher No F-00612 consortium was from the collection of D.K. Zabolotny, Institute of Microbiology and Virology, National Academy of Sciences, Ukraine. *Pseudomonas laurentiana* (strain DSM ID 20-163) was isolated from *Salvia officinalis* roots and identified by the DSMZ (Braunschweig, Germany). The species has a 100% similarity to *Pseudomonas laurentiana* GLS-010 (MG719526.1). The *Arthrobacter* species doi:10.13145/bacdive131619.20201210.5, subsequently named MPI 764, and doi:10.13145/bacdive131618.20201210.5, subsequently named MPI 763, and *Pseudomonas spec*. MPI 9 (NCBI taxonomic identity 1736604) were from P. Schulze-Lefert, Max Planck Institute for Plant Breeding Research, Cologne, Germany.

#### 3.5.2. Incubations of Microorganisms

The incubation of the microorganisms was similar to the method described in Schütz et al. [17]. *Aminobacter aminovorans*, *Paenibacillus polymyxa*, *Bacillus amyloliquefaciens*, *Pantoea ananatis*, *Bacillus amyloliquefaciens*, *Pseudomonas* spec. MPI 9, and the *Arthrobacter* species MPI 763 and MPI 764 were precultured in LB medium for 2 days. Then, 1 mL of the precultures was transferred to flasks with 15 mL Czapek medium containing 1 mg of **5** or 1 mg of **2** (the latter only for *Aminobacter aminovorans*, *Paenibacillus polymyxa*, *Pseudomonas* spec. MPI 9, and the *Arthrobacter* species MPI 763 and MPI 764) and cultured for 3 days up to 1 week in the dark without shaking. The fungus *Papiliotrema baii* was precultured in YEP medium for 3 days, then 1 mL of the cultures were cultivated in 15 mL Czapek medium with 1 mg **5** for 1 week. The consortia were grown on Sabouraud agar for 3 (*A. elegans*) and 7 days (*T. viride*). For incubations, 0.5 cm^2^ agar plugs with mycelium were placed into flasks containing 15 mL Czapek medium and 2 mg of **5** and incubated for 7 days. All incubations were run at 25 °C in the dark without shaking. Next, 200 µL aliquots of the incubation media were taken after 1, 3, and 7 days. After the incubation period at 25 °C, the cultures were placed in a cold room (4 °C) and further monitored by HPLC for compounds up to three weeks by taking 200 µL aliquots of the culture media every 3 days (HPLC/diode array detector (DAD), Shimadzu [51], detection range 200–800 nm). *Aminobacter aminovorans* and *Paenibacillus polymyxa* were also grown in Czapek medium with vigorous shaking during the first 2 days and then further cultured in the dark without shaking.

*Pantoea ananatis*, *Pseudomonas laurentiana*, and *Papiliotrema baii* were incubated with 500 µL of the concentrated glucosylated **5** from *Arabidopsis* in Czapek and LB medium, as described above. Aliquots (200 µL) of the media were taken after 24 h, 48 h, and after additional 7 days. The aliquots were checked by HPLC for hydrolysis of **9** (Appendix A).

#### 3.5.3. Extraction of Culture Media and Microorganisms

After incubation, the liquid cultures (media plus the microorganisms: total cultures) were extracted with ethyl acetate and the organic and aqueous phases evaporated in vacuo. The residues were dissolved in methanol, and the solutions were centrifuged for 15 min at 20,000 rpm and 4 °C. The supernatants were analyzed for new compounds by HPLC using the abovementioned method.

The supernatants from the *Aminobacter aminovorans* and *Paenibacillus polymyxa* cultures were immediately dried in a Speedvac vacuum concentrator before storing at −80 °C. Aliquots were directly checked by HPLC for the successful conservation of the new compounds seen before. The compositions of the crude extracts were consistent with the data obtained later after reconstitution of the dry extracts with CH_3_CN/MeOH, showing the same four main peaks (*P1-4*) in HPLC/UV runs at 254 nm (Appendix A). Due to the instability of some of the products, no further purifications were performed prior to the subsequent isolations and analyses.

#### 3.5.4. Plants

*Brassica oleracea* var. *gongylodes* L. (kohlrabi) plants were grown hydroponically on cheese cloth for 7 days, as described. Surface sterilized *Arabidopsis thaliana* Col-0 seeds were germinated and grown on MS Phytoagar for three weeks in a phytochamber under a long-daylight regime (6 h light/8 h dark at 18 °C).

#### 3.5.5. Plant Incubation, Extraction of Roots, and Leaves of Plants

Twenty kohlrabi or *Arabidopsis* seedlings were placed on Petri dishes (4 dishes/incubation experiment) containing 50 mL of water-soluble **5** (0.5 mM). Kohlrabi was also incubated with **7** and **8**. The seedlings were incubated for 72 h, and afterwards they were intensively washed; then, the roots were cut, dried, weighted, and homogenized with 50% methanol (0.5 mL/100 mg roots). The homogenates were immediately vortexed for 1 min and centrifuged for 5 min, and the supernatants were either directly analyzed by HPLC or stored at –20 °C until analysis.

For the identification of derivatives of **5** or of **7**, the homogenates prepared from cut roots of seedlings from six kohlrabi incubations with **5** or with **7** were evaporated to dryness, the residues were dissolved in 50% methanol, and the resulting suspension was centrifuged at 20,000 rpm at 4 °C for 5 min. The supernatants were analyzed by HPLC, intensively dried, and stored at −80 °C until they were used for compound identification.

Detection of glucosylated **5** in *Arabidopsis* roots was performed after incubation of the plants with 3 mg of **5**/10 mL water for 72 h. The residue left after evaporation of the crude extract was dissolved in H_2_O, the suspension centrifuged to remove insoluble material and the supernatant was evaporated in vacuo at 70 °C to moistness. The moist residue was treated twice with pure methanol, centrifuged and the supernatant dried again, dissolved in 50% methanol, and centrifuged for 15 min at 20,000 rpm and 4 °C. The final supernatant contained glucosylated **5** as the major product. To support the structure elucidation, 20 µL aliquots of the concentrated fraction were incubated with 1 mU *β*-glucosidase from almonds (Sigma-Merck) for 30 min in 200 µL 100 mM acetate buffer pH 5.5. The reaction was stopped by boiling, the mixture was centrifuged for 5 min at 20,000 rpm and 4 °C, and the supernatant was analyzed by HPLC (Appendix A). Analysis of the assay mixture after incubation with the *β-*glucosidase showed complete hydrolysis of **9** to **5**, thus confirming the identity of the substrate as compound **9**.

3-week-old *Arabidopsis* plants were incubated at room temperature under natural light conditions for 48 h with 0, 0.3, 0.5, 1, 2 and 3 mM of **5** in Petri dishes to estimate effects on plant growth. Incubation (12 h) with 0.5 mM nitrated BOA-6-OH (**12**) was performed for comparison [17]. The plants were washed and extracted, as described above, but with 1 mL 50% methanol/100 mg plant material. The extracts were analyzed by HPLC, and the content of metabolites **5** and **9** were determined using external standard curves (detection wavelength 254 nm).

### 3.6. Analyses of Compounds in Microbial Culture Media and Plant Extracts

#### 3.6.1. Sample Preparation

The crude dry bacterial extracts were reconstituted with CH_3_CN/MeOH 3:1 using vortexing/sonication. After centrifugation, the supernatant was filtered using a Socorex^®^ borosilicate glass syringe and 0.2 μm MilliQ Millipore^®^ LCR filters. An aliquot of the filtrate was 10x diluted with MeOH prior to injection (1 μL).

#### 3.6.2. LC/HRMS Methods

Chromatographic separations were performed on a *Dionex*™ *UltiMate**™ 3000* UHPLC system (Thermo Fisher Scientific, Waltham, MA, USA), equipped with a CTC PAL autosampler, DAD detector, heated-column compartment, and solvent-delivery system. On-line coupled QExactive™ Orbitrap^®^ mass spectrometer (Thermo Fisher Scientific, USA), featuring heated electrospray ionization (HESI II^®^) ion source and higher energy dissociation collision cell (N_2_ as collision gas), was used for acquisition of the MS and MS^2^ data. Mass accuracy was calibrated below 2 ppm using Pierce^®^ LTQ Velos calibration solution, in accordance with the requirements of the manufacturer. All operations were controlled by Xcalibur software (v.4.2.28.14, Thermo Scientific, USA). The ion-source tune settings were set to default for the specified flow rate. LC conditions (method 1 and 2) and MS parameters can be consulted in Appendix A.

#### 3.6.3. isCID/MS^2^ Experiments in ESI Mode

To optimize the fragmentation behavior of target analytes, a test solution of **5** (200 µg/mL MeOH) was injected *on-flow* (450 µL/min, 23%B), using a T-connector and syringe pump at 10 µL/min flow rate.

#### 3.6.4. NMR Measurements

The 2D NMR data of the collected fractions were acquired with a Bruker AV-600 spectrometer, equipped with a BOSS-II shim system, a digital lock control unit, AMOS Control system, DQD unit, BVT3000/BCU05 cooling unit, and Cryo platform. CRYO TCI inverse triple resonance (^1^H; ^13^C; ^15^N) probe (5 mm) was employed for the acquisition. The synthetic compounds were measured on a Bruker AV-500 spectrometer equipped with 5 mm BB (^1^H; ^13^C) probe; chemical shifts (*∂* in ppm) were calibrated with the signal of DMSO-d_6_ (∂ = 2.50 ppm in ^1^H- and 39.52 ppm in ^13^C-NMR experiments); and coupling constants were *J* in Hz.

#### 3.6.5. Semi-Preparative Purification of the Major Compounds

The extract obtained after incubation of *A. aminovorans* with 2-acetamido-phenol was used for the identification of the three main products. The methanolic solution reconstituted from the dried sample was fractionated on a Shimadzu HPLC system consisting of two LC20-AP solvent delivery units, SPD-20A UV/Vis detector, equipped with an analytical UV cell, CBM-20A control module, and FRC-10A fraction collector (Shimadzu, Kyoto, Japan). A Cortecs™ C18 (4.6 × 150 mm, 2.7 μm, 90Å) HPLC column (Waters^®^, Milford, MA, USA) was employed for all separations at room temperature. The mobile phase consisted of H_2_O + 0.1% *v*/*v* HCOOH (A) and CH_3_CN + 0.1% *v*/*v* HCOOH (B). The linear gradient used for the fraction collection started at 0–2.5 min 5% B, then 2.5–10.5 min 5–30% B, 10.5–11.5 min 30–95% B, 11.5–15.5 min 95% B, 15.5–17 min 95–5% B, and 17–25 min 5% B. Flow rate was kept constant at 1 mL/min. Temperature of the UV/Vis analytical cell was maintained at 30 °C. The fractions were collected automatically according to UV signal (254 nm). Eight consecutive collection runs were performed with compulsory retention time control after each separation. Solutions were collected in *Eppendorf*^®^ vials (5 mL) and evaporated to dryness under N_2_ stream at 30 °C, and the residues were reconstituted with a small amount of MeOH. The solvent was again evaporated under N_2_ stream at 30 °C, with DMSO-d_6_ immediately added thereafter. In order to obtain sufficient material for structural elucidation with NMR spectroscopy, separation and purification of the main components were repeated eight times, and the resulting fractions were pooled. The purities and purification yield of the samples of interest, *P2*, *P3*, and *P4*, were determined after evaporation of the solvent (*P2*, >70%, 832 µg, *P3*, >70%, 600 µg, and *P4*, >90%, 440 µg). These overall yields are only approximate values due to partial sample decomposition after solvent evaporation (color of the residue changed rapidly from yellow to orange/red on contact with air).

The total *Aminobacter aminovorans* culture extract *F49* contained additional 2-acetamido-phenol derivatives, which were analyzed by extracted ion chromatograms of the LC-(–)-ESI-HRMS and identified as dimeric compounds: six dimers, five nitrosylated dimers, and five nitrated dimers.

The semi-preparative purification of the fraction at 12.5 min (Appendix A), by the method described above, disclosed several further derivatives, which were subjected to LC-(–)-ESI-HRMS analysis, revealing the existence of several trimeric compounds (see Figure 5 and Appendix A).

### 3.7. Real-Time PCR

Three-week-old *Arabidopsis* plants were incubated with 1 mM of **5** for 0 (control), 30 min, 1, 3, 5, and 24 h. Then, 100 mg of the plants from each incubation time and from the corresponding control plants incubated without substrate **5** were harvested and immediately frozen in liquid nitrogen until RNA was extracted. The material was homogenized with the Precellys^®^ 24/24-Dual (PEQLAB Biotechnology), and RNA was isolated with the NucleoSpin RNA Plant Kit (Macherey and Nagel, Germany), in accordance with the instructions of the manufacturer. For cDNA preparation, the Thermo Scientific RevertAid First Strand cDNA Synthesis Kit was used. RT-PCR was performed with a 7300 real time PCR system (Applied Biosystems) for studying the relative expression of *VTE1* (tocopherol cyclase) [52], *TPS04* and *TPS02* (terpene synthases) [53], *TRYPS02* (tryptophan synthase beta-subunit 1) [54], *GLN1.1* and *GLN1.2* (glutamine synthases) [35], and *NIA1* and *NIA2* (nitrate reductases *NR1*, *NR2*) [55]. The reaction mixture (20 µL) consisted of 4 µL dd H_2_O, 5 µL cDNA, 1 µL primer pair (working solution 10 pmol/µL), and 10 µL 5xEvaGreen (ROX) qPCR-Mix II (Bio-Budget Technologies, Krefeld, Germany). Primers are listed in Appendix A. Gene expression was determined relative to the control plants and to actin (*ACT2*), which was used as reference gene. For evaluation of RT-qPCR results, the ΔΔCT method was applied.

## Figures and Tables

**Figure 1 molecules-27-04786-f001:**
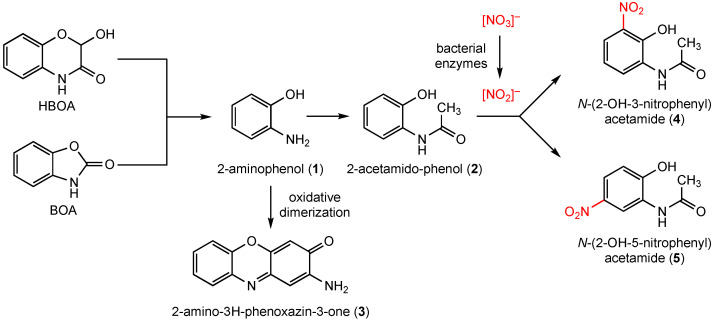
Microbial degradation of benzoxazolin-2(*3H*)-one (BOA) and the benzoxazolinone derivative HBOA. The degradation product 2-aminophenol (**1**) is converted to 2-acetamidophenol (**2**), the substrate for microbial nitration at position 3 and 5. In another route, **1** is oxidatively dimerized to 2-amino-3H-phenoxazin-3-one (**3**), a substrate for numerous substitution reactions by microorganisms [3,4].

**Figure 2 molecules-27-04786-f002:**
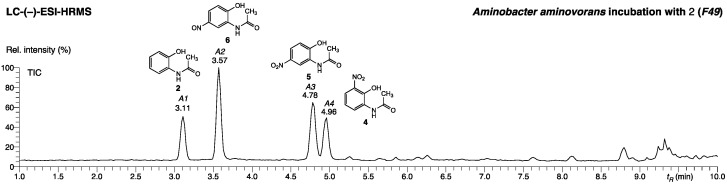
LC-(–)-ESI-HRMS chromatogram (Waters Acquity BEH 100 × 2 mm, 1.7 µm column) of *F**49* (*Aminobacter aminovorans)* from total culture extract after incubation with **2**. Chromatographic peaks *A1-A4* belong to **2** and its conversion products **4**, **5**, and **6**.

**Figure 3 molecules-27-04786-f003:**
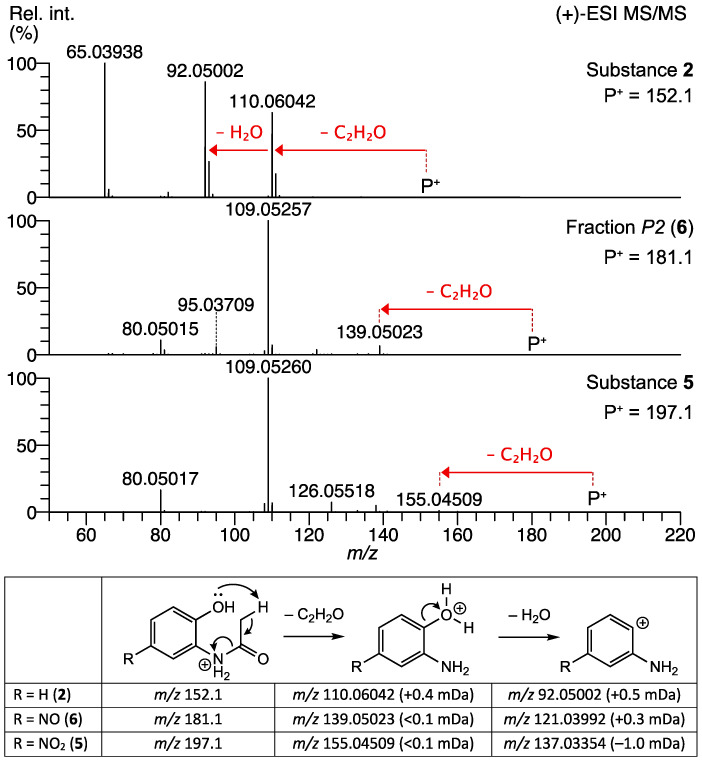
(+)-ESI-MS/MS spectra of *N*-(2-hydroxy-5-nitrosophenyl)acetamide (**6**) from fraction *P2* of extract *F49* produced by *A. aminovorans* (middle) in comparison with the commercially available compounds **2** (top) and **5** (bottom).

**Figure 4 molecules-27-04786-f004:**
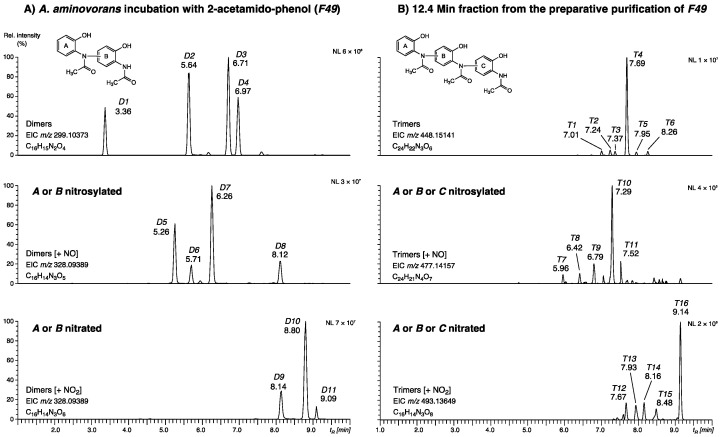
High-resolution extracted-ion chromatograms (EIC, ±1.5 mDa) obtained from the LC-(–)-ESI-HRMS analysis of (**A**) fraction *F49*, revealing dimers of 2-acetamido-phenol (**A**: peaks *D1*-*D11*) and their nitrosylated and nitrated derivatives and (**B**) fraction *t_R_* 12.5 min of purified *F49* (Appendix A), indicating the presence of analogous trimers (**B**: peaks *T1*-*T16*).

**Figure 5 molecules-27-04786-f005:**
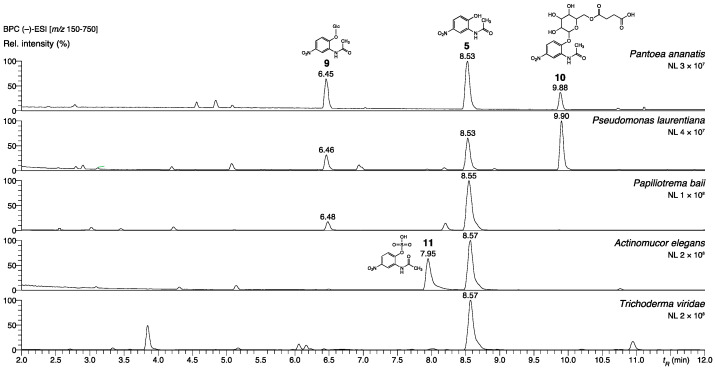
LC-(–)-ESI-HRMS base peak chromatograms with annotated derivatives of compound **5** in total bacterial or fungal culture extracts after incubation with **5**. Analytical data can be consulted in Appendix A.

**Figure 6 molecules-27-04786-f006:**
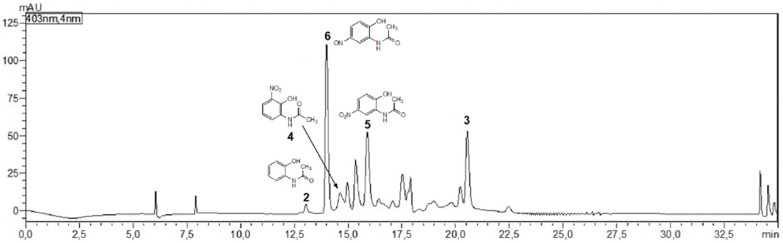
LC-UV chromatogram (403 nm) of an *A. aminovorans* culture supplemented with **2** and shaken during the first 2 days of culture. The medium contained some 2-amino-*3H*-phenoxazin-3-one (**3**) as well as **2**, **4**, **5**, and **6**.

**Figure 7 molecules-27-04786-f007:**
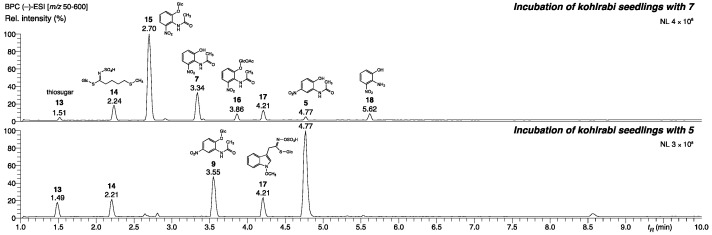
LC-(–)-ESI-HRMS chromatograms of kohlrabi root exudate extracts after incubation with **5** (bottom) and **7** (top). MS/MS data and annotation of the metabolites are presented in Appendix A.

**Figure 8 molecules-27-04786-f008:**
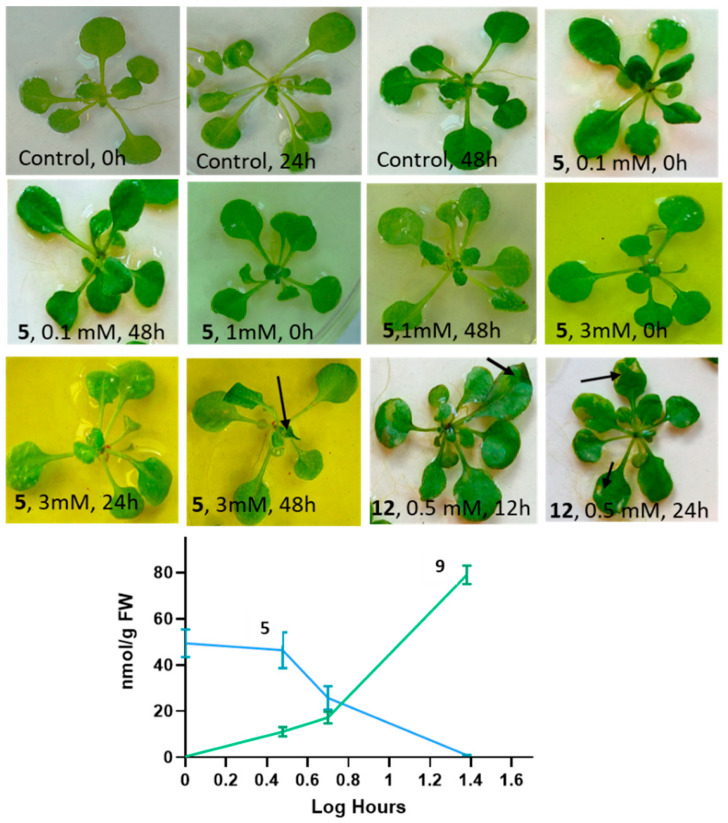
Phenotype of *Arabidopsis* plants exposed to **5** for 0, 24, and 48 h in comparison to 0.5/1.0 mM 6-hydroxy-5-nitrobenzo[*d*]oxazol-2(3*H*)-one (**12**) (incubation times 12 and 24 h). Overall, **12** severely damaged the leaves (arrows). With **5**, only the highest concentration (3 mM) led to wilting of young leaves after 48 h (arrow). The graph shows the decrease in **5** (blue line) and accumulation of its glucosylated derivative **9** (green line) in *Arabidopsis* plants after 1, 3, 5, and 24 h of incubation with 1 mM **5**.

**Figure 9 molecules-27-04786-f009:**
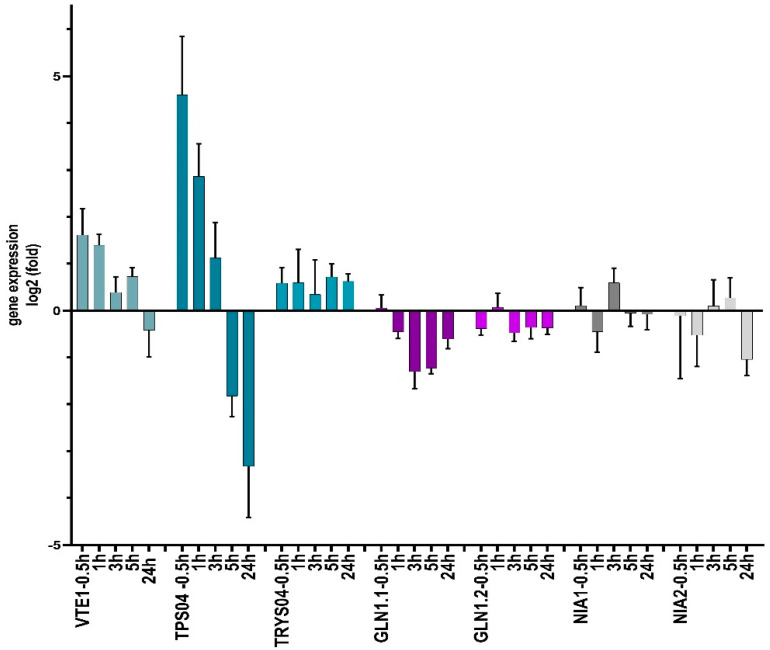
Incubation of 3-week-old *Arabidopsis thaliana* plants with 1 mM of substrate **5**. Time dependent relative expression levels (log2 ^–ΔΔCt^) of *VTE1* (tocopherol cyclase), *TPS04* and *TPS02* (terpene synthases), *TRYPS02* (tryptophan synthase beta-subunit 1), *GLN1.1* and *GLN1.2* (glutamine synthases), and *NIA1* and *NIA2* (nitrate reductases *NR1*, *NR2*). Means ± SD are pictured. Primers are shown in Appendix A.

**Figure 10 molecules-27-04786-f010:**
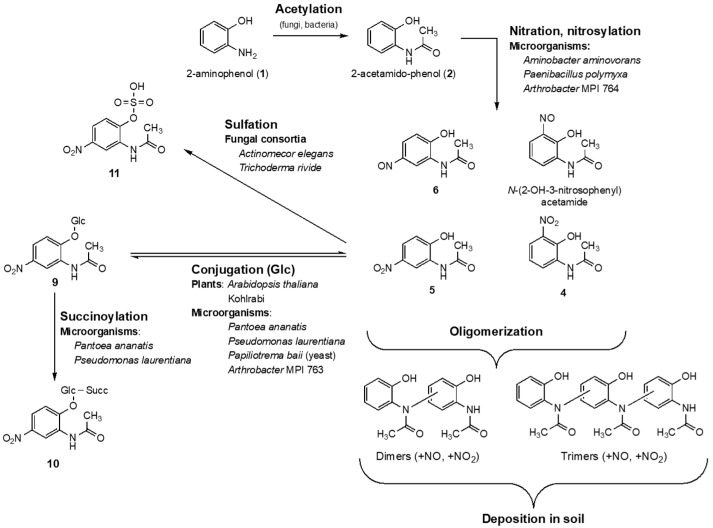
Alternative biotransformation of 2-aminophenol (**1**) by microorganisms and fungi in the presence of nitrate in soil.

**Table 1 molecules-27-04786-t001:** Summary of compounds identified in total bacterial culture extracts obtained from *A. aminovorans* and *P. polymyxa* after incubation with **2**. The (–)-ESI mass spectra are shown in Appendix A. The identification levels are based on the guidelines of Metabolomics Standards Initiative [18,19,20,21].

Peak	*t_R_* [min]	Compound	Structure	(–)-ESI (*m*/*z*)Deviation [mDa]	ID Level	Reference
*A1*	3.11	2-Acetamido-phenol (**2**)	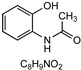	150.056210.2	1	Commercially available
*A2*	3.57	*N*-(2-Hydroxy-5-nitrosophenyl)acetamide (**6**)	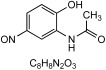	179.046370.2	2	New structure
*A3*	4.78	*N*-(2-Hydroxy-5-nitrophenyl)acetamide (**5**)	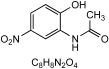	195.041340.2	1	Commercially available
*A4*	4.96	*N*-(2-Hydroxy-3-nitrophenyl)acetamide (**4**)	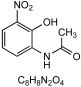	195.041360.2	1	Synthesized compound

## Data Availability

Not applicable.

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
