# Peer review of "Bioactive Nitrosylated and Nitrated N-(2-hydroxyphenyl)acetamides and Derived Oligomers: An Alternative Pathway to 2-Amidophenol-Derived Phytotoxic Metabolites"

_molecules, 2022, doi:10.3390/molecules27154786_

Round 1

Reviewer 1 Report

The presentation of the information in the article is very clear and above all gives an explanation of each of the results obtained. In general, the explanation of the proposed 2-aminophenol compound pathway is complete and clear.

Check figures 3 and 4, and that the text at the top of the figure is cut, if read, but could correct.

Check bibliography 4, as there is extra space before the first name.

Author Response

We thank the reviewers for their valuable comments that helped us to improve the manuscript.  Following we give our statements to the reviewers’ comments. We hope we considered all points of criticism and the manuscript can be accepted for publication.

1- Overall this manuscript is acceptable, as it expands the current knowledge of the metabolism of BOA/HBOA, and thus is of interest to a plant sciences niche. Disappointingly, this manuscript is not written for audiences outside this niche, specially, the much broader natural products chemistry readership (why is a combination of compound names and abbreviations inconsistently used throughout the text, why not compound numbers as is traditional in natural products chemistry?). 

Changed. No abbreviations are presented anymore, although the agronomists and biologists working with benzoxazinoids are familiar with the abbreviations but probably not the exact chemical names of the compounds. Numbers change with the publications. Except for chemists, all others will have problems now with the text. The niche is smaller than before.

Although most microbial experiments appear to have been carefully planned and executed, the plant feeding experiments seem problematic (employing isotopically labeled compounds would appear necessary). Furthermore, the objectives, and some sections of the discussion and conclusions are confusing. Some omissions ask for a "leap of faith" from the readers, e.g., "phenoxazine" is part of the title and mentioned in the text as an important entity, but no chemical structure or explanation of its importance is given. 

Phenoxazinone was part of the title because we present an alternative pathway to phenoxazinone production which is the topic of our work. The generation of phenoxazinones is explained in the introduction. The new title lacks the word phenoxazinone.

Specific issues (by no means a comprehensive list):

1- The title of this submission is too long and somewhat confusing, please simplify/clarify (e.g., Bioactive N-(2-Hydroxy-2/5-Nitrophenyl)acetamides and Derived Oligomers: An Alternative Pathway to 2-Amidophenol Derived Phytotoxic Metabolites).
Why does the graphical abstract include only "soil bacteria" when there is a lot more work reported in this manuscript?  

Title is changed.

The graphical abstract can only illustrate the most important part of the work but not all aspects. The major part of our manuscript are compounds synthesized by bacteria. In the instructions for authors it is written:  « it (the GA) should represent the topic of the article ». But we add some additional informations.

2- Introduction:
Figure 1 - essential corrections: please number o-aminophenol as 1 and change accordingly all other numbers; figure states incorrect names of compounds 3 and 4, please renumber and use correct names(delete the word "amides"). Please insert pertinent refs. in legend since this is a literature synopsis.

Changed. We included refs. [3, 4].

Please state objectives clearly. 
What is the general chemical structure of phenoxazine(s)? How relevant is it in this work?

The structure is given in the revised Fig. 1. In the part Implication of Nitration of 2-Acetamido-phenol  it is written: The alternative pathway for 2-aminophenol derived nitro-acetamido-phenol compounds can reduce the formation of phenoxazinone, a compound assumed to have potential as a bioherbicide.We think the relevance of phenoxazinones is clear. The new pathways demonstrate how phenoxazinone generation can be bypassed and this point is the major finding of the presented work.

Compound numbers must be used throughout the text including Materials and Methods, please avoid letters such as AP, AAP, or APO (in this context it is not suitable to use letters and numbers, such a mix makes the textual arguments more difficult to follow). 

See above.

3- Results and Discussion:
Figs. that show LC-chromatograms, namely, Figs. 2, 5, 6, 7, must also include the chromatograms of corresponding controls, so that the reader is able to assess the validity of all data.

Done, shown in the supplementary figures.

Fig. 8 is poorly organized, difficult to follow A, B, C.

Done. Fig. 8 was newly organized.

Please make the discussion clearer, specially making clearer that a relevant conclusion(s) is not a speculation.

Text passages, possibly addressed by the reviewer (not clear), were rewritten.

Important addition: in the overall conclusion, include a new metabolic scheme with pathway(s) that display the name of organism/species analyzed and the corresponding chemical structures of all isolated and/or identified metabolites. New Figure 10. shows a summary of the new metabolic pathways.

Reviewer 2 Report

The publication submitted for review is well written and deserves to be published. I have no questions for the authors.

Author Response

Reviewer 2 had no questions to the authors.